# Spread them Apart: Towards Robust Watermarking of Generated Content

## Abstract

Generative models that can produce realistic images have improved significantly in recent years. The quality of the generated content has increased drastically, so sometimes it is very difficult to distinguish between the real images and the generated ones. Such an improvement comes at a price of ethical concerns about the usage of the generative models: the users of generative models can improperly claim ownership of the generated content protected by a license. In this paper, we propose an approach to embed watermarks into the generated content to allow future detection of the generated content and identification of the user who generated it. The watermark is embedded during the inference of the model, so the proposed approach does not require the retraining of the latter. We prove that watermarks embedded are guaranteed to be robust against additive perturbations of a bounded magnitude. We apply our method to watermark diffusion models and show that it matches state-of-the-art watermarking schemes in terms of robustness to different types of synthetic watermark removal attacks.

## 1 Introduction

Recent advances in generative models have brought the performance of image synthesis tasks to a whole new level. For example, the quality of the images generated by diffusion models [DMs, Croitoru et al. (2023); Rombach et al. (2022); Esser et al. (2024)] is now sometimes comparable to the one of the human-generated pictures or photographs. Compared to generative adversarial networks [GANs, Goodfellow et al. (2014); Brock et al. (2019)], diffusion models allow the generation of high-resolution, naturally looking pictures and incorporate much more stable training, leading to more diverse generation. More than that, the image generation process with diffusion models is more stable, controllable, and explainable. They are easy to use and are widely deployed as tools for data generation, image editing [Kawar et al. (2023); Yang et al. (2023)], music generation [Schneider et al. (2024)], text-to-image synthesis [Saharia et al. (2022); Zhang et al. (2023); Ruiz et al. (2023)] and in other multimodal settings.

Unfortunately, there are several ethical and legal issues that may arise from the usage of diffusion models. On the one hand, since diffusion models can be used to generate fake content, for example, deepfakes [Zhao et al. (2021); Narayan et al. (2023)], it is crucial to develop automatic tools to verify that a particular digital asset is artificially generated. On the other hand, a dishonest user of the model protected by a copyright license can query it, receive the result of generation, and later claim exclusive copyright. In this work, we focus on the detection of the content generated by a particular model and the identification of the end-user who queried the model to generate a particular content. We develop a technique to embed the digital watermark into the generated content during the inference of the generative model, so it does not require retraining or fine-tuning the generative model. The approach allows not only to verify that the content was generated by a source model but also to identify the user who sent a corresponding query to the generative model. We prove that the watermark embedded is robust against additive perturbations of the content of a bounded magnitude.

Our contributions are threefold:

- We propose *Spread them Apart*, the framework to embed digital watermarks into the generative content of continuous nature. Our method embeds the watermark during the process

Figure 1: Illustration of the proposed method. During the image generation phase, the user $u_i$ queries the model with the prompt. Given the prompt, the model produces the latent $z$, from which the image is generated. If the image generated satisfies the constraint $\mathcal{L}_{wm} < \varepsilon$ (meaning the watermark is successfully embedded), it is yielded to the user; otherwise, the loss function from equation 10 is minimized with the respect to the latent $z$. Note that the value of $\varepsilon$ may vary from image to image. During the watermark retrieval phase, given the image $x$ and $m$ secrets, $s(u_1), \ldots, s(u_m)$, the watermark decoder extracts m watermarks, $w(u_1|x), \ldots, w(u_m|x)$. Then, the image is attributed to the user $u$ according to the equation 9.

of content generation and, hence, does not require additional training of the generative model.

- We apply the framework to watermark images generated by a diffusion model and prove that the watermark embedded is provably robust to the additive perturbations of a bounded magnitude that can be applied during the post-processing of the image.

- Experimentally, we show that our approach outperforms competitors in terms of the robustness to different types of post-processing of the images aimed at watermark removal, such as brightness and contrast adjustment or gamma correction.

## 2 RELATED WORK

### 2.1 DIFFUSION MODEL

Inspired by non-equilibrium statistical physics, [Sohl-Dickstein et al. (2015)] introduced the diffusion model to fit complex probability distributions.[Ho et al. (2020)] introduced a new class of models called Denoising Diffusion Probabilistic Models (DDPM) by establishing a novel connection between the diffusion model and the denoising scoring matching. Later, the Latent Diffusion Model (LDM) [Rombach et al. (2022)] was developed to improve efficiency and reduce computational complexity, with the diffusion process happening within a latent space $\mathcal{Z}$. During training the LDM uses an encoder $\mathcal{E}$ to map an input image $x$ to the latent space: $z = \mathcal{E}(x)$. For the reverse operation a decoder $\mathcal{D}$ is employed, so that $x = \mathcal{D}(z)$. During inference, the LDM starts with a noise vector $z \sim \mathcal{N}(0, I)$ in the latent space and iteratively denoises it. The decoder then maps the final latent representation back to the image space.

### 2.2 WATERMARKING OF DIGITAL CONTENT

Watermarking has been recently adopted to protect the intellectual property of neural networks [Wu et al. (2020); Pautov et al. (2024)] and generated content [Kirchenbauer et al. (2023); Zhao et al. (2024); Fu et al. (2024)]. In a nutshell, watermarking of generated content is done by injection of digital information within the generated image allowing the subsequent extraction. Existing methods of digital content watermarking can be divided into two categories: content-level watermarking and model-level watermarking. The methods of content-level watermarking operate in some representation of content, for example, in the frequency domain of the image signal [ó Ruanaidh et al. (1996); Cox et al. (1996)]. When the image is manipulated in the frequency domain, the watermark embedding process can be adapted to produce watermarks that are robust to geometrical image

transformations, such as rotations and translations [Wen et al. (2024)]. Model-level watermarking approaches are designed to embed information during the generation process. In end-to-end methods, the models to embed and extract watermark are learned jointly [Zhu et al. (2018); Hayes & Danezis (2017)]. In [Yu et al. (2021)], it was proposed to teach the watermark encoder on the training data of the generative model; such an approach yields a watermarking scheme that is conditioned on the generative model and its training dataset. This method was later adapted to latent diffusion models [Fernandez et al. (2023)] and unconditional diffusion models [Zhao et al. (2023)]. In contrast, there are methods that do not require additional model training. These methods are designed to alter the output distribution of the generative model to embed previously learned watermark into the model or the content itself [Kirchenbauer et al. (2023); Wen et al. (2024)].

### 2.3 ROBUSTNESS TO WATERMARK REMOVAL ATTACKS

Watermarking attacks are aimed at removing the watermark embedded into the model's weights or generated content. In the prior works on removing the watermarks from generated images [Li et al. (2019); Cao et al. (2019)], the attack problem is formulated in terms of the image-to-image translation task, and methods to remove watermarks via an auxiliary generative adversarial network are presented. Other approaches [Hertz et al. (2019); Liang et al. (2021); Sun et al. (2023)] perform watermark removal in two steps: firstly, the visual watermark is localized within an image; secondly, it is removed via a multi-task learning framework.

In practice, watermarking scheme has to be robust to destructive and constructive attacks, or synthetic transformations of the data. Destructive transformations, such as brightness and contrast adjustment, geometric transformations, such as rotations and translations, compression methods, and additive noise are aimed at watermark removal by applying a transformation. In contrast, constructive attacks treat watermarks as noise and are aimed at the restoration of original content [Zhang et al. (2024)]. It is usually done by applying purification techniques, such as Gaussian blur [Hosam (2019)] or image inpainting [Liu et al. (2021); Xu et al. (2017)].

Signal Processing Attacks focus on noise addition, compression, and filtering. Robust watermarking schemes based on frequency domain transformations and randomizing offered higher resilience against these types of attacks Taran et al. (2019).

## 3 PROBLEM STATEMENT

In this section, we formulate the problem statement and the research objectives. Note that we focus on the watermarking of images generated by diffusion models, but the formulation below is valid for watermarking of any generated content, for example, audio, video, or text.

### 3.1 IMAGE WATERMARKING

In our approach, we focus on *detection* and *attribution* of the generated image simultaneously: while detection is aimed to verify whether a particular image is generated by a given model, attribution is aimed at determining the user who generated the image.

Suppose that we are given the generative model $f$ deployed in the black-box setting, i.e., as a service: in the generation phase a user $u_i \in [u_1, \ldots, u_m]$ sends a query to the model and receives a generated image $x \in \mathbb{R}^d$. If $x$ is a watermarked image, the owner of model $f$ should be able to identify that $x$ is generated by user $u_i$ by querying the model $f$. In our method, the image is watermarked during the *generation* phase, not during the post-processing. We formulate the process of watermarking and attribution in the following way:

1. When the user $u_i \in [u_1, \ldots, u_m]$ registers in the service, it is assigned a pair of *public* and *private* keys, namely, the watermark $w(u_i)$ and the secret $s(u_i)$. Watermark is a binary string of length $n$ and the secret is the sequence of tuples of length $n$, where each tuple is a pair of unique positive numbers treated as indices: $w(u_i) \in \{0, 1\}^n$, $s(u_i) \in \mathbb{Z}_+^{2n}$.

2. When the user $u_i$ queries the model $f$, it generates the image $x$ with the watermark $w(u_i)$ embedded in it.

3. When the watermarked object $x$ is received by the model owner, it extracts the watermark $w(u_i|x)$ using the secret $s(u_i)$ of the user $u_i$ from it and compares it with the watermark $w(u_i)$ assigned to the user $u_i$. Following the previous works [Yu et al. (2021); Fernandez et al. (2023)], we compute the bitwise distance $d(w(u_i|x), w(u_i))$ between $w(u_i|x)$ and $w(u_i)$:

$$d(w(u_i|x), w(u_i)) = \sum_{j=1}^{n} \mathbb{1}(w(u_i|x)_j \neq w(u_i)_j). \tag{1}$$

**Remark.** *For the purposes of robustness to watermark removal attack, in case of a single user $u_i$, we flag the object $x$ as generated by the user $u_i$ if the distance $d(w(u_i|x), w(u_i))$ is either small or large, namely, if*

$$d(w(u_i|x), w(u_i)) \in [0, \tau_1] \cup [\tau_2, n], \tag{2}$$

*where $\tau_1 \ll n$ and $\tau_2 \gg 0$. This procedure is known as the double-tail detection [Jiang et al. (2023)].*

## 3.2 The Probability of Incorrect Attribution

We assume that the watermark $w(u_i)$ attributed to the user $u_i$ is drawn randomly and uniformly from the set of all possible $n-$bit watermarks, $\{0,1\}^n$. Following the prior works [Fernandez et al. (2023)], we formulate the detection problem as the hypothesis test. In case of a single user $u_i$, we define the null hypothesis $\mathcal{H}_0 = $ "the object $x$ is generated not by $u_i$" and the alternative hypothesis $\mathcal{H}_1 = $ "the object $x$ is generated by $u_i$". Additionally, under the null hypothesis, we assume that the $j'$th bit in the watermark $w(u_i|x)$ extracted from $x$ is the same as the $j'$th bit from $w(u_i)$ with the probability $p_i$

In the case of a single user $u_i$ and given the attribution rule from the Equation 2, we compute the probability of the false attribution, namely,

$$FRP(1)|_{u_1} = \mathbb{P}_{w' \sim \{0,1\}^n, w' \neq w(u_i)} [d(w', w(u_i)) \in [0, \tau_1] \cup [\tau_2, n]] =$$

$$\sum_{q \in [1, \tau_1] \cup [\tau_2, n]} \binom{n}{q} p_i^q (1 - p_i)^{n-q}, \tag{3}$$

where $w' = w(u_i|x)$.

In case of $m$ users, the probability $FPR(m)$ of incorrect attribution of the non-watermarked image $x$ to some other user $u_j \in [u_1, \ldots, u_m]$ is upper bounded by the probability below:

$$FPR(m) \leq \mathbb{P}_{w' \sim \{0,1\}^n} [\exists u_j \in [u_1, \ldots, u_m] : d(w', w(u_j)) \in [0, \tau_1] \cup [\tau_2, n]] \leq$$

$$\leq \sum_{u_j \in [u_1, \ldots, u_m]} FPR(1)|_{u_j} = \hat{p}. \tag{4}$$

Note that this upper bound holds regardless of the independence of random variables $\xi_1, \ldots, \xi_m$, where

$$\xi_i = \mathbb{1}[d(w(u_i|x), w(u_i)) \in [0, \tau_1] \cup [\tau_2, n]]. \tag{5}$$

**Remark.** *In our experiments, the probability $p_i$ from above is estimated to be close to $\frac{1}{2}$.*

## 3.3 Robustness to Watermark Removal Attacks

When the user $u_i$ receives the watermarked image $x$, it can post-process it to obtain the other image, $x'$, which does retain the sufficient part of the watermark $w(u_i)$. The transition from $x$ to $x'$ may be done by applying an image transformation, such as brightness or contrast adjustment, Gaussian blur, or additive noise. The other approach is to perform an adversarial attack on the generative model to erase the watermark [Jiang et al. (2024)]. In our settings, we assume that the generative model is deployed as the black-box service with limited access to the API, so an adversary can not apply white-box adversarial attacks [Jiang et al. (2023)].

# 4 METHOD

In this section, we provide a detailed description of the proposed approach, its implementation details, and the robustness guarantee against additive watermarking removal attacks of bounded magnitude.

## 4.1 SPREAD THEM APART: EMBEDDING AND EXTRACTION OF THE WATERMARK

Suppose that $f$ is the generative model. Recall that the user $u_i \in [u_1, \ldots, u_m]$ receives a pair $(w(u_i), s(u_i))$ after the registration in the service, where both the watermark and the secret are unknown to the user and are privately kept by the owner of $f$. Let $x$ be the generated image. Then, the watermark embedding process is described as follows:

1. The secret $s(u_i)$ is interpreted as two sequences of indices, $A = \{a_1, \ldots, a_n\}$ and $B = \{b_1, \ldots, b_n\}$. The watermark $w(u_i) = \{w_1, \ldots, w_n\}$ is the binary string that restricts the generated image $x$ in the areas represented by the sets $A$ and $B$.

2. The restriction of $x$ in the areas represented by the sets $A$ and $B$ given $w(u_i)$ is the following implication:

$$\begin{cases} w_i = 0 \implies x_{a_i} \geq x_{b_i} \\ w_i = 1 \implies x_{a_i} < x_{b_i}, \end{cases} \quad (6)$$

where $x_j$ is the intensity of the $j'$th pixel of $x$. To increase the robustness to watermark removal attacks, we apply additional regularization to $x$:

$$\min_{j \in [1, \ldots, n]} |x_{a_j} - x_{b_j}| \geq \epsilon, \quad (7)$$

where $\epsilon > 0$ is the scalar parameter.

To perform detection and attribution of the given image $x$, the owner of the generative model firstly constructs $m$ watermarks $w(u_1|x), \ldots, w(u_m|x)$ by reversing the implication from the Equation 6. Namely, given the secret $s(u_i) = \{a_1, \ldots, a_n, b_1, \ldots, b_n\}$ of user $u_i$, the watermark bits are restored by the following rule:

$$\begin{cases} x_{a_j} \geq x_{b_j} \implies w(u_i|x)_j = 0, \\ x_{a_j} < x_{b_j} \implies w(u_i|x)_j = 1. \end{cases} \quad (8)$$

**Remark.** *Here, we distinguish the watermark $w(u_i)$ assigned by the owner of generative model to the user $u_i$ from the watermark $w(u_i|x)$ extracted from the image $x$ with the use of the secret $s(u_i)$ of user $u_i$.*

When $m$ watermarks $w(u_1|x), \ldots, w(u_m|x)$ are extracted, the owner of the model assigns $x$ to the user $u$ with the minimum distance $d(w(u_i), w(u_i|x))$ between assigned and extracted watermarks:

$$u = \arg \min_{u_i \in [u_1, \ldots, u_m]: \, \xi_i = 1} d(w(u_i), w(u_i|x)), \quad (9)$$

where $\xi_i$ is the indicator function from the Equation 5. Note that if $\xi_i = 0$ for all $i \in [1, \ldots, m]$, then $x$ is identified as image not generated by $f$.

## 4.2 SPREAD THEM APART: IMPLEMENTATION DETAILS

In this subsection, we describe the watermarking procedure. First of all, we have to note that in the Stable Diffusion model, the latent vector $z$ produced by the U-Net is then decoded back into the image space using a VAE decoder: $x = \mathcal{D}(z)$. To embed the watermark into an image, we optimize a special two-component loss function with respect to the latent vector $z$. The overall loss is written as follows:

$$\mathcal{L} = \lambda_{wm}\mathcal{L}_{wm} + \lambda_{qual}\mathcal{L}_{qual}, \quad (10)$$

The first term, $\mathcal{L}_{wm}$, defines how the image complies with the pixel difference imposed by the watermark $w(u_i) = \{w_1, \ldots, w_n\}$ and the secret $s(u_i) = \{a_1, \ldots, a_n, b_1, \ldots, b_n\}$:

$$\mathcal{L}_{wm} = \sum_{i=1}^{n} \min((-1)^{w_i}(x_{a_i} - x_{b_i}) + \varepsilon, 0), \quad x = \mathcal{D}(z), \tag{11}$$

Here, $\varepsilon$ defines the minimum difference between private key pixels that we would like to obtain. Note that the larger the value of $\varepsilon$ is, the more robust the watermark is to additive perturbations. At the same time, the increase of $\varepsilon$ negatively influences the perceptual quality of images.

The second term $\mathcal{L}_{qual}$, is introduced to preserve the generation quality of the image. The value $\mathcal{L}_{qual}$ is difference in image quality measured by LPIPS metric Zhang et al. (2018), that acts as a regularization. Given $x$ and $y$ as the input images, the LPIPS metric is defined as follows [Ghazanfari et al. (2023)]:

$$d(x, y) = \sum_{j} \frac{1}{W_j H_j} \sum_{w,h} \|\phi^j(x) - \phi^j(y)\|_2^2. \tag{12}$$

Here, $\phi^j(x) = w_j \odot o_{hw}^j(x)$, where $o^j(x)$ are the internal activations of the CNN, AlexNet [Krizhevsky et al. (2012)], in our case.

Note that we do not perform denoising at each iteration, as we only manipulate the latent vectors produced by U-Net; the forward step of the described optimization procedure involves only the decoding of the latent vectors: $x = \mathcal{D}(z)$.

The optimization is performed over 700 steps of the Adam optimizer with the learning rate of $8 \times 10^{-3}$, where every 100 iteration, the learning rate is halved. When the convergence is reached, the ordinary Stable Diffusion post-processing of the image is performed. The coefficients $\lambda_{wm}$ and $\lambda_{qual}$ are determined experimentally and set to be 0.9 and 150, respectively, the value of $\varepsilon$ was set to be $\varepsilon = 0.2$. Schematically, the process of watermark embedding and extraction is presented in Figure 1.

## 4.3 SPREAD THEM APART: ROBUSTNESS GUARANTEE

By construction, the watermark embedded by our method is robust against additive watermark removal attacks of a bounded magnitude. Namely, let the watermark $w(u_i|x)$ be embedded in $x$ with the use of the secret $s(u_i) = \{a_1, \ldots, a_n, b_1, \ldots, b_n\}$ of the user $u_i$. Let

$$\Delta_i = \frac{|x_{a_i} - x_{b_i}|}{2}. \tag{13}$$

Then, the following lemma holds.

**Lemma 4.1.** *Let $\varepsilon \in \mathbb{R}^d$ and $\Delta_{i_1} \leq \Delta_{i_2} \leq \cdots \leq \Delta_{i_n}$.*

*Then, if $\|\varepsilon\|_\infty < \Delta_{i_k}$, then $d(w(u_i|x + \varepsilon), w(u_i|x)) < k$.*

*Proof.* Note that to change the $j'$th bit of watermark $w(u_i|x)$, an adversary has to change the sign in expression $(x_{a_j} - x_{b_j})$. Without the loss of generality, let $x_{a_j} - x_{b_j} \geq 0$.

Consider an additive noise $\varepsilon$ such that $(x + \varepsilon)_{a_j} - (x + \varepsilon)_{b_j} < 0$, meaning $|\varepsilon_{b_j} - \varepsilon_{a_j}| > |x_{a_j} - x_{b_j}|$. Note that $\|\varepsilon\|_\infty \geq \max(|\varepsilon_{a_j}|, |\varepsilon_{b_j}|)$.

If $\max(|\varepsilon_{a_j}|, |\varepsilon_{b_j}|) < \Delta_j$, then $|\varepsilon_{b_j} - \varepsilon_{a_j}| \leq |\varepsilon_{b_j}| + |\varepsilon_{a_j}| < 2\Delta_j = |x_{a_j} - x_{b_j}|$, yielding a contradiction. Thus, $\|\varepsilon\|_\infty \geq \Delta_j$.

Finally, an observation that all the indices in $s(u_i)$ are unique finalizes the proof. $\square$

This lemma provides a lower bound on the $l_\infty$ norm of the additive perturbation $\varepsilon$ applied to $x$ which is able to erase at least $k$ bits of the watermark $w(u_i|x))$ embedded in $x$.

## 5 EXPERIMENTS

### 5.1 GENERAL SETUP

For the experiments, we use `stable-diffusion-2-base` model [Rombach et al. (2022)] with the `epsilon` prediction type and 50 steps of denoising. The resolution of generated images is

$512 \times 512$. The experiments were conducted on `DiffusionDB` dataset [Wang et al. (2022)]. Specifically, we choose 1000 unique prompts and generate 1000 different images.

The public key for the user is sampled from the Bernoulli distribution with the parameter $p = 0.5$. The length of a key is set to be $n = 100$. The private key is generated by randomly picking $2n$ unique pairs of indices of the flattened image.

## 5.2 ATTACK DETAILS

We evaluate the robustness of the watermarks embedded by our method against the following watermark removal attacks: brightness adjustment, contrast shift, gamma correction, image sharpening, hue adjustment, saturation adjustment, random additive noise, JPEG compression, and the white-box PGD attack adversarial [Madry et al. (2018)]. In this section, we describe these attacks in detail.

Brightness adjustment of an image x was performed by adding a constant value to each pixel: $x_{brightness} = x + b$, where $b$ was sampled from the uniform distribution $\mathcal{U}[-20, 20]$.

Contrast shift was done in two ways: positive and negative. The positive contrast shift implies the multiplication of each pixel of an image by a constant positive factor: $x_{contrast} = cx$, where $c$ was sampled from the uniform distribution, $c \sim \mathcal{U}[0.5, 2]$.

In contrary, when the contrast shift is performed with the negative value of $c$ (namely, $c \sim \mathcal{U}[-2, -0.5]$), such a transform turns an image into a negative. Later, we treat these transforms separately and denote them as "Contrast $+$" and "Contrast $-$", depending on the sign of $c$.

Gamma correction is nothing but taking the exponent of each pixel of the image: $x_{gamma} = x^g$, where $g \sim \mathcal{U}[0.5, 2]$.

For sharpening, hue, and saturation adjustment, we use implementations from the `Kornia` package [Riba et al. (2020)] with the following parameters: $a_{saturation} = 2.0$, $a_{hue} = 0.2$ and $a_{sharpness} = 2.0$.

The noise for the noising attack was sampled from the uniform distribution $\mathcal{U}[-\delta, \delta]$, where $\delta$ was chosen to be 25. Note, that the maximum $\|\cdot\|_{\infty}$ of noise is then equal to 25.

JPEG compression was performed by means of DiffJPEG [Shin (2017)] with quality equal to 50.

White-box attack aims to change the embedded watermark $w$ to some other watermark $\tilde{w}$ by optimizing the image with respect to the loss initially used to embed the watermark $w$:

$$\mathcal{L}_{wb} = \lambda_{wm}\mathcal{L}_{wm} + \lambda_{qual}\mathcal{L}_{qual}, \quad \mathcal{L}_{wm} = \sum_{i=1}^{n} \min((-1)^{\tilde{w}_i}(x_{a_i} - x_{b_i}) + \varepsilon, 0). \quad (14)$$

In equation 14, the term $\mathcal{L}_{qual}$ corresponds to the difference in image quality in terms of LPIPS metric, namely,

$$\mathcal{L}_{qual} = LPIPS(x, \hat{x}), \quad (15)$$

where $x$ and $\hat{x}$ are the original image and image on a particular optimization iteration, respectively.

The loss function $\mathcal{L}_{wb}$ pushes the private key pixels to be aligned with a new randomly sampled public key $\tilde{w}$, so that the ground-truth watermark $w$ gets erased. The attack's budget is the upper bound of $\|\cdot\|_{\infty}$ norm of the additive perturbation, that we have taken to be $\varepsilon/2$ from the equation 11. Let $\tilde{x}$ be the image obtained after the attack. If at some iteration the distance between the source image $x$ and the attacked one $\tilde{x}$ exceeds $\varepsilon/2$, $\tilde{x}$ is being projected back onto the sphere $\|\tilde{x} - x\|_{\infty} = \varepsilon/2$. The optimization took place for 10 iterations with the Adam optimizer and the learning rate was equal to $10^{-1}$. Note that this attack setting implies knowledge about the private key and assumes white-box access to the generative model. Hence, this is de facto the strongest watermark removal attack we consider.

Pixels of the images perturbed by the attacks are then linearly mapped to $[0, 255]$ segment:

$$x^{(i)} = 255 \frac{x^{(i)} - x^{(i)}_{min}}{x^{(i)}_{max} - x^{(i)}_{min}}, \quad i \in \{R, G, B\}. \quad (16)$$

Table 1: Image quality metrics. The best results are highlighted in **bold**.

| Metric | Stable Signature | AquaLora | WOUAF | Ours |
|---|---|---|---|---|
| SSIM ↑ | 0.89 | **0.92** | — | 0.86 |
| PSNR ↑ | **30.0** | 29.42 | — | 29.4 |
| FID ↓ | 19.6 | 24.72 | > 15.0 | **13.2** |
| LPIPS ↓ | — | — | — | 0.0072 |

## 5.3 RESULTS

In this section, we provide the quantitative results of experiments. We report (i) quality metrics of the generated images (SSIM, PSNR, FID and LPIPS) to evaluate the invisibility of the watermarks, (ii) bit-wise error of the watermark extraction caused by watermark removal attacks and (iii) True Positive Rates in attribution and detection problems.

We compare our results (where applicable) to that of Stable Signature Fernandez et al. (2023), SSL watermarking Fernandez et al. (2022), AquaLora Feng et al. (2024) and WOUAF Kim et al. (2024), one of the state-of-the-art watermarking approaches. In these works the watermark length is set to be 48, 30, 48 and 32, respectively, while we have 100 bits long watermarks: note that the longer the watermark, the harder it is to be embedded.

The image quality metrics are presented in the Table 1. It can be seen that our results are comparable to the ones of the baseline methods in terms of the quality of produced images and significantly surpass them in terms of the FID metric. Qualitative comparison of original and watermarked images can be found in Figure 2. More examples are provided in Appendix A.2.

To evaluate the robustness of the watermarks against removal attacks, we report an average bit-wise error, ABWE:

$$ABWE = \frac{1}{N_{images} \times n} \sum_{i=1}^{N_{images}} \sum_{j=1}^{n} \mathbb{1}[w_{i,j}^{gt} \neq w_{i,j}^{extracted}], \tag{17}$$

where $w_{i,j}^{gt}$ and $w_{i,j}^{extracted}$ are the $j$-th bits of ground truth and extracted private keys, corresponding to the $i$-th image. Here, $n$ is the number of bits in the watermark. We report ABWE in the Table 2.

To estimate the TPR in the attribution problem, we extract $k = 10$ different watermarks from the watermarked images. To extract a different watermark, we randomly generate $k = 10$ different private keys to simulate other users. The results are reported in Table 3 together with the TPRs under different watermark removal attacks. Note that the PGD attack in this setting is aimed at restoring the original watermark. To estimate the TRP in the watermark detection problem, we do the same procedure for non-watermarked images generated by the Stable Diffusion model and extract $k = 10$ different watermarks. The results are presented in the Table 4.

Note that our framework yield both low misattribution and misdetection rates according to the two-tail detection and attribution rules from the equation 9.

## 5.4 LIMITATIONS

Note that the proposed approach has several limitations. First of all, since the watermarking is performed during the model's inference, it affects both the inference time and, in some cases, the quality of the generated images: the watermarked images can have artifacts in contrast to their non-watermarked counterparts. See Fig. 5 in Appendix for details. Note that these artifacts, although visible, barely spoil the images' content. Secondly, the proposed watermarking method does not provide robustness against cropping, rotation, and translation attacks. However, the robustness to rotation and translation can be achieved by inserting the watermarks in the frequency domain of the image.

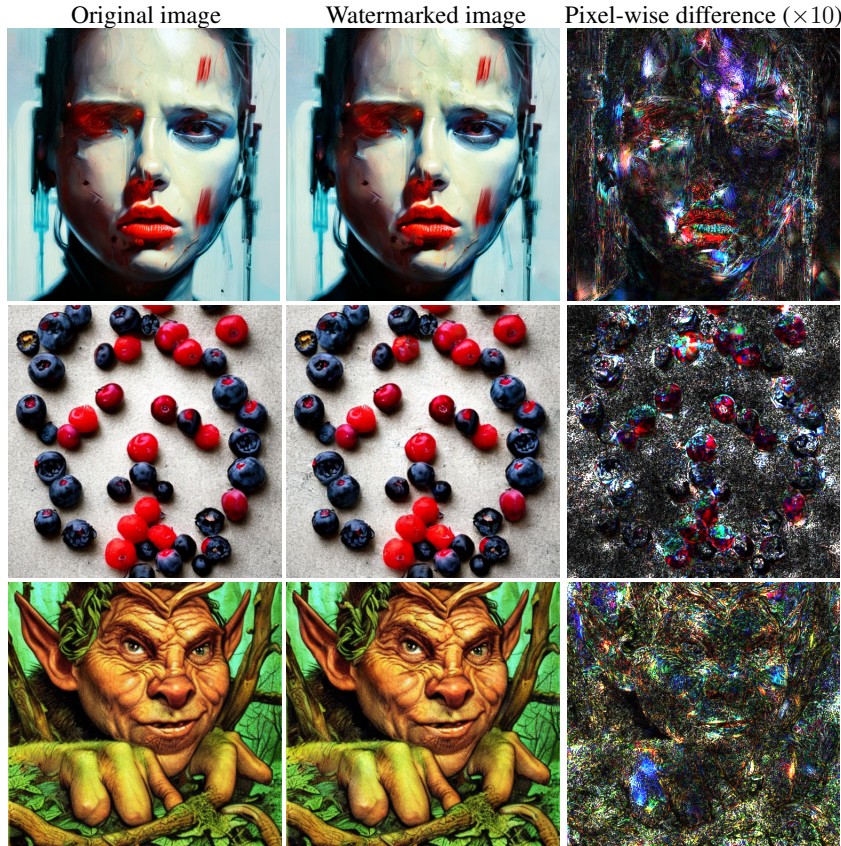

Original image · Watermarked image · Pixel-wise difference (×10)

Figure 2: Examples of watermarked images. The maps of absolute pixel-wise difference between source images and the generated ones were added for the illustration purposes.

Table 2: Average bit-wise error after watermark removal attacks. The column "Generation" corresponds to the average bit-wise error of the watermarking process itself. The best results are highlighted in **bold**.

| Method | Generation | Brightness | Contrast + | Contrast − | Gamma | JPEG |
|---|---|---|---|---|---|---|
| Ours | 0.0008 | **0.002** | **0.002** | **0.998** | **0.003** | 0.147 |
| Stable signature | 0.01 | 0.03 | 0.02 | — | — | 0.12 |
| SSL watermarking | **0.00** | 0.06 | 0.04 | — | — | **0.04** |
| AquaLora | 0.0721 | — | — | — | — | 0.0508 |

| Method | Hue | Saturation | Sharpness | Noise | PGD |
|---|---|---|---|---|---|
| Ours | **0.01** | 0.1 | **0.0008** | **0.057** | **0.064** |
| Stable signature | — | **0.01** | 0.01 | — | — |
| SSL watermarking | 0.06 | — | — | — | — |
| AquaLora | — | — | — | 0.07 | |

## 6 CONCLUSION

In this paper, we propose *Spread them Apart*, the framework to watermark generated content of continuous nature and apply it to images generated by Stable Diffusion. We prove that the watermarks produced by our method are provably robust against additive watermark removal attacks of a bounded norm. Our approach can be used to both detect that the image is generated by a given model and to identify the end-user who generated it. Experimentally, we show that our method is

Table 3: TPRs under different types of watermark removal attacks, attribution problem. We use $k = 10$ different private keys and fix FPR = $10^{-6}$. Such a FPR is achieved when $\tau_1 = 19$ and $\tau_2 = 81$ from equation 5. The parameters of removal attacks are presented in Section 5.2. The best results are highlighted in **bold**.

| Method | Generation | Brightness | Contrast $+$ | Contrast $-$ | Gamma | JPEG |
|---|---|---|---|---|---|---|
| Ours | **1.000** | **1.000** | **1.000** | **1.000** | **1.000** | 0.444 |
| Stable signature | 0.998 | 0.927 | — | — | — | 0.784 |
| AquaLora | 0.998 | — | — | — | — | **0.998** |
| WOUAF | **1.000** | 0.997 | — | — | — | 0.969 |

| Method | Hue | Saturation | Sharpness | Noise | PGD |
|---|---|---|---|---|---|
| Ours | **1.000** | **0.653** | **1.000** | 0.971 | **0.862** |
| Stable signature | — | — | — | 0.776 | 0.747 |
| AquaLora | — | — | — | 0.958 | — |
| WOUAF | — | — | — | **0.982** | — |

Table 4: TPRs under different types of watermark removal attacks, detection problem. We use $k = 10$ different private keys and fix FPR = $10^{-6}$. Such a FPR is achieved when $\tau_1 = 19$ and $\tau_2 = 81$ from equation 5. The parameters of removal attacks are presented in Section 5.2.

| Method | Generation | Brightness | Contrast $+$ | Contrast $-$ | Gamma | JPEG |
|---|---|---|---|---|---|---|
| Ours | **1.000** | **1.000** | **1.000** | **1.000** | **1.000** | 0.444 |
| Stable signature | **1.000** | 0.862 | — | — | — | 0.217 |
| SSL watermarking | 1.000 | 0.940 | 0.960 | — | — | **0.810** |

| Method | Hue | Saturation | Sharpness | Noise | PGD |
|---|---|---|---|---|---|
| Ours | **1.000** | **0.653** | **1.000** | **0.971** | **0.862** |
| Stable signature | — | — | — | 0.406 | 0.505 |
| SSL watermarking | **1.000** | — | — | — | — |

comparable to the state-of-the-art watermarking methods in terms of the invisibility of watermark and the robustness to synthetic watermark removal attacks.

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

## A  APPENDIX

### A.1  ADDITIONAL EXPERIMENTS

In this section, we provide the results of additional experiments. Namely, we provide the evaluation of time cost of our method, additional ablation experiments, comparison with other baselines, and discuss an extensions of our approach to provide watermark robustness to geometric transformations, such as rotation and translation.

Table 5: Average time in seconds required to embed a watermark.

| Method | Watermark embedding time, sec. |
|---|---|
| Ours | 35.7 |
| Stable Signature | $\approx 60.0$ |
| SSL watermarking | — |
| AquaLora | $\approx 0.0$ |
| WOUAF | 1.1 |

### A.1.1 COMPUTATIONAL COST

Recall that the proposed method implies an auxiliary optimization procedure during the inference of the model. In Table 5, we report time in seconds required to generate a watermarked image and compare it to that of the other methods.

### A.1.2 SCALABILITY OF THE METHOD

Note that the watermark extraction procedure implies the comparison of the extracted watermark, given the private key, with the public keys of the users. Namely, to extract the watermark, one should pass the private key $s(u_i)$ of user $u_i$ and compare extracted watermark $w(u_i x)$ with the watermark $w(u_i)$ assigned to $u_i$. In Table 6, we report the average time of watermark extraction, depending on the number $m$ of users in the database.

Table 6: Time in seconds required to extract a watermark, depending on the number $m$ of users in the database. All the experiments were conducted on a single GPU Nvidia H100, time is averaged over 100 executions.

| $m$ | 1 | 10 | 1000 | 10000 | 1000000 |
|---|---|---|---|---|---|
| Time, sec. | $7.5 \times 10^{-5}$ | $7.4 \times 10^{-4}$ | $7.2 \times 10^{-2}$ | $6.9 \times 10^{-1}$ | 71.2 |

### A.1.3 ABLATION STUDY

Note that both the robustness of watermark to image transformations and quality of generated images depend on the parameters of experiments. To choose the best combination of parameters in terms of trade-off between the robustness and image quality, one can perform ablation study.

In Tables 7-8, we report quantitative results of ablation study. In each table, we report the values of the varying parameter, while leaving the default values of other parameters (namely, $n = 100, \varepsilon = 0.2, \lambda_{wm} = 0.9, \lambda_{qual} = 150$).

### A.1.4 ROBUSTNESS TO GEOMETRIC TRANSFORMATIONS

Recall that our method does not provide the provable robustness against geometric transformations, such as rotations and translations, out-of-the-box. However, slight modification of our method can be done to achieve robustness to rotations and translations. Namely, one can embed a watermark not into pixels of an image, but into the corresponding invariant in the Fourier space [Lin & Brandt (1993)]:

**Theorem A.1.** *Suppose $f(x, y)$ is an integrable nonnegative function and its Fourier transform $F(\omega_x, \omega_y)$ is differentiable at the origin. Then the following complex function, called the phase Taylor invariant,*

$$T(\omega_x, \omega_y) = F(\omega_x, \omega_y)e^{-i(a\omega_x + b\omega_y)}, \tag{18}$$

*where*

$$a = -i\frac{|F(0,0)|}{F(0,0)} \frac{\partial}{\partial \omega_x} \frac{F(\omega_x, \omega_y)}{|F(\omega_x, \omega_y)|}(0,0) \quad and \quad b = -i\frac{|F(0,0)|}{F(0,0)} \frac{\partial}{\partial \omega_y} \frac{F(\omega_x, \omega_y)}{|F(\omega_x, \omega_y)|}(0,0) \tag{19}$$

Table 7: Ablation study: the effect of the parameter values on the robustness of watermark. We report average bit-wise error and study the robustness to JPEG, Hue, Saturation, Sharpness and Gaussian noise, since our approach provide robustness to brightness, contrast and gamma shifts by design. Default settings are colored by gray cells.

| Parameter | Value | JPEG | Hue | Saturation | Sharpness | Noise |
|---|---|---|---|---|---|---|
| $n$ | 50 | 0.123 | 0.013 | 0.095 | 0.002 | 0.049 |
| | 100 | 0.143 | 0.011 | 0.104 | 0.001 | 0.056 |
| | 150 | 0.157 | 0.013 | 0.112 | 0.001 | 0.063 |
| | 250 | 0.159 | 0.015 | 0.120 | 0.001 | 0.069 |
| $\varepsilon$ | 0.0 | 0.313 | 0.109 | 0.206 | 0.016 | 0.202 |
| | 0.05 | 0.261 | 0.055 | 0.169 | 0.005 | 0.159 |
| | 0.2 | 0.143 | 0.011 | 0.104 | 0.001 | 0.056 |
| | 0.5 | 0.054 | 0.001 | 0.041 | 0.000 | 0.003 |
| $\lambda_{wm}$ | 0.5 | 0.150 | 0.015 | 0.108 | 0.002 | 0.060 |
| | 0.9 | 0.143 | 0.011 | 0.104 | 0.001 | 0.056 |
| | 2.0 | 0.136 | 0.012 | 0.103 | 0.001 | 0.056 |
| $\lambda_{qual}$ | 10.0 | 0.059 | 0.014 | 0.071 | 0.004 | 0.035 |
| | 50.0 | 0.088 | 0.008 | 0.082 | 0.001 | 0.040 |
| | 150.0 | 0.143 | 0.011 | 0.104 | 0.001 | 0.056 |
| | 200.0 | 0.160 | 0.013 | 0.109 | 0.001 | 0.060 |

Table 8: Ablation study: the effect of the parameter values on the image quality. We report the values of SSIM, PSNR, LPIPS image quality metrics. In the first column, we report the varying parameter. Default settings are colored by gray cells.

| Parameter | Value | SSIM ↑ | PSNR ↑ | LPIPS ↓ |
|---|---|---|---|---|
| $n$ | 50 | 0.897 | 31.104 | 0.006 |
| | 100 | 0.856 | 29.381 | 0.007 |
| | 150 | 0.827 | 28.309 | 0.009 |
| | 250 | 0.777 | 26.726 | 0.013 |
| $\varepsilon$ | 0.0 | 0.878 | 30.142 | 0.006 |
| | 0.05 | 0.873 | 29.937 | 0.007 |
| | 0.2 | 0.856 | 29.381 | 0.007 |
| | 0.5 | 0.820 | 28.378 | 0.010 |
| $\lambda_{wm}$ | 0.5 | 0.869 | 29.830 | 0.006 |
| | 0.9 | 0.856 | 29.381 | 0.007 |
| | 2.0 | 0.842 | 28.912 | 0.008 |
| $\lambda_{qual}$ | 10.0 | 0.752 | 26.200 | 0.057 |
| | 50.0 | 0.806 | 27.601 | 0.019 |
| | 150.0 | 0.856 | 29.381 | 0.007 |
| | 200.0 | 0.869 | 29.918 | 0.005 |

*is invariant under translation.*

**Theorem A.2.** *Let $\tilde{f}(r,t) = f(e^r \cos t, e^r \sin t)$ be the change of coordinates to the logarithmic-polar ones. Denote Fourier-Mellin transform of $\tilde{f}(r,t)$ as*

$$\tilde{F}(\omega, k) = \int_{-\infty}^{\infty} \int_{0}^{2\pi} \tilde{f}(r,t) e^{-i(kt+\omega r)} dt dr = \tilde{A}(\omega, k) e^{-i\tilde{\psi}(\omega, k)}, \quad (20)$$

| Original image | Rotated image | Translated image |

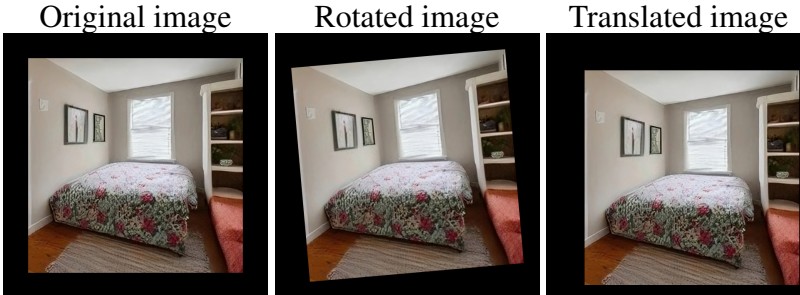

Figure 3: Examples of geometric transformations.

Table 9: TPRs under geometric transformations, JPEG, cropping and erasing, detection problem. We set FPR = $10^{-6}$.

| Method | Rot. | Trans. | JPEG (50) | Crop ($400 \times 400$) | Erase ($160 \times 160$) |
|---|---|---|---|---|---|
| Ours (Fourier) | 0.850 | 1.000 | 0.700 | 0.800 | 0.900 |
| Stable sign. | 0.970 | — | 0.880 | 0.988 | — |
| SSL | 1.000 | — | 0.970 | 1.000 | — |
| AquaLora | — | — | 0.998 | 0.919 | — |
| WOUAF | 0.990 | — | 0.971 | 0.988 | 0.990 |

*where $\tilde{A}(\omega, k)$ is the magnitude and $\tilde{\psi}(\omega, k)$ is the phase. Then, $\tilde{A}(\omega_x, \omega_y)$ is invariant under rotation.*

Note that for Theorems A.1-A.2 to hold, geometric transformations should be done without the loss of information (i.e., rotation and translation on an infinite plane) Lin & Brandt (1993). To emulate such transformations, we firstly pad images before rotating and translating them. In Fig. 3, examples of these transformations are presented.

In Table 9, we report the robustness of our updated approach (denoted as "Ours (Fourier)") to geometric transformations and JPEG compression and compare the results with the other baselines.

## A.2 QUALITATIVE RESULTS

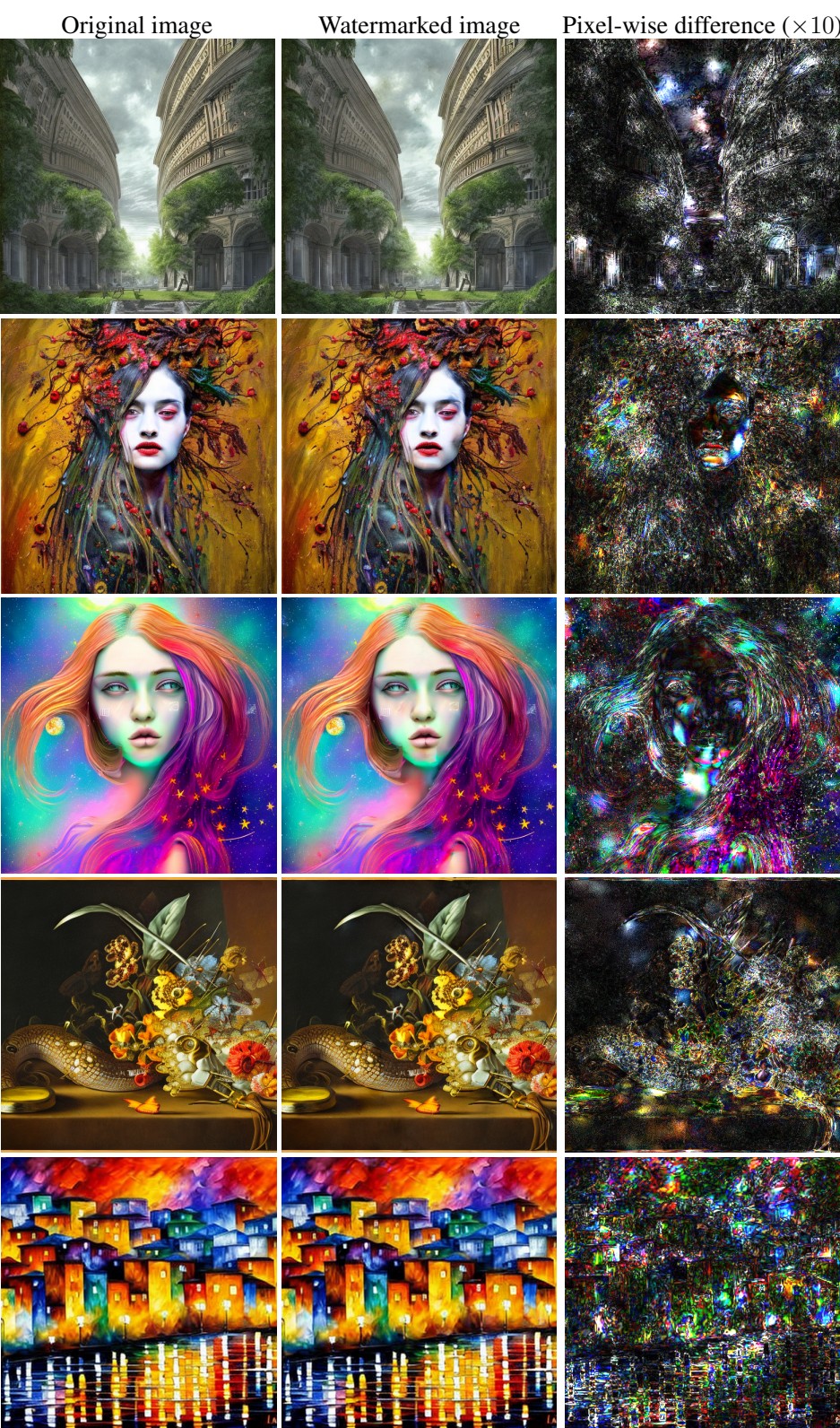

Figure 4: Additional examples of watermarked images with ×10 pixel-wise difference with the original images.

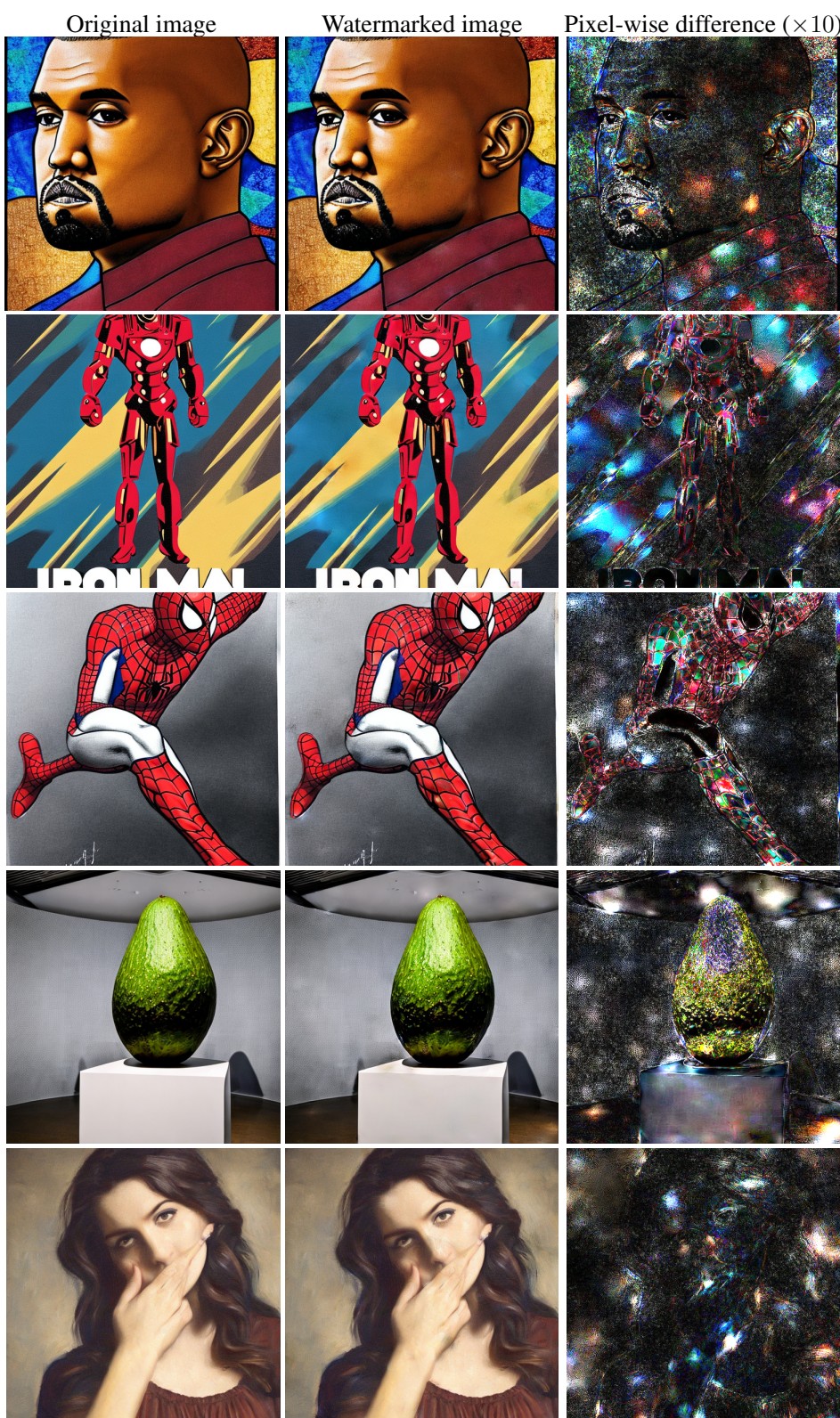

Figure 5: Examples of watermarked images with artifacts.

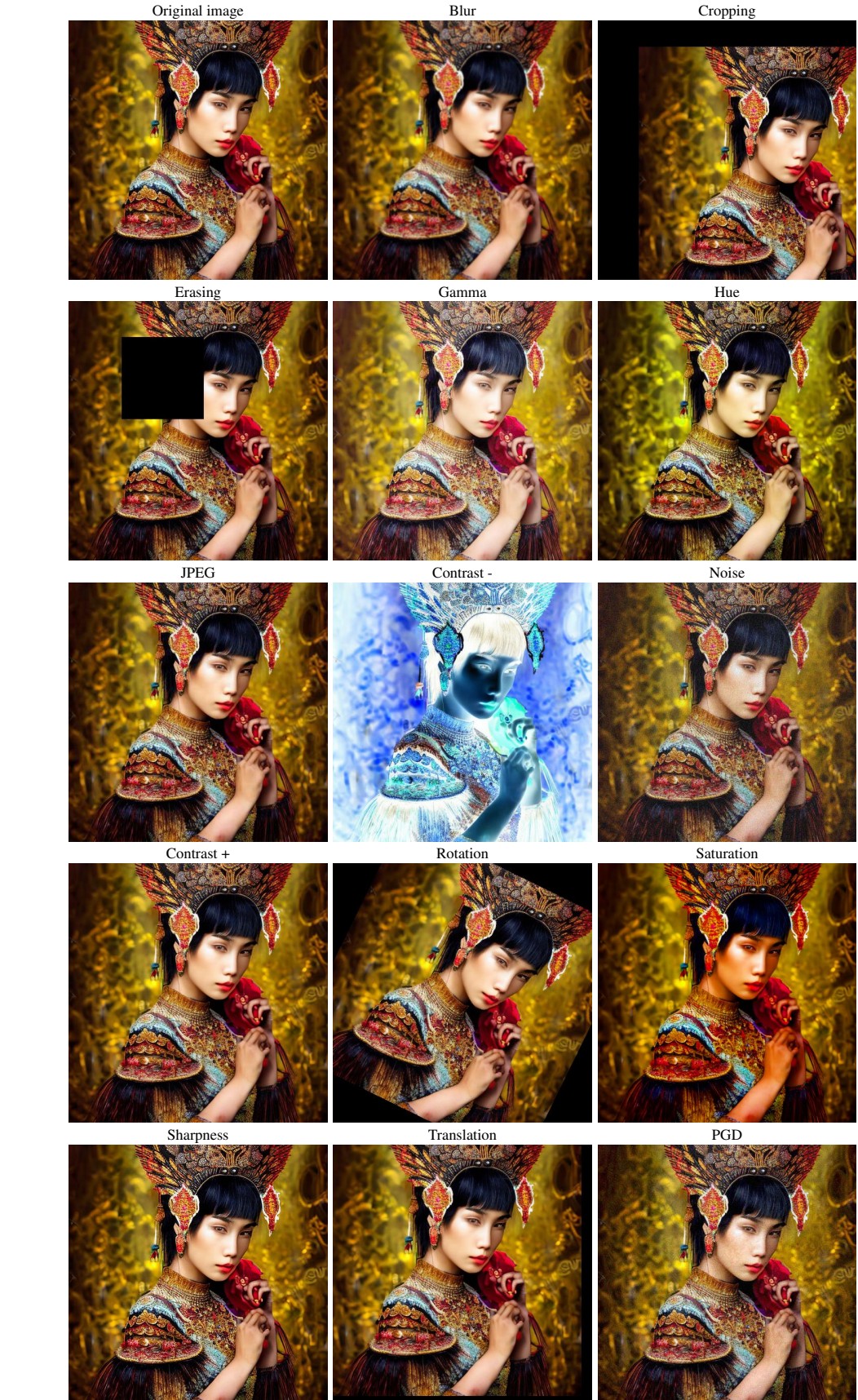

Figure 6: Examples of corrupted images.

| Original image | Watermarked image | Pixel-wise difference ($\times 10$) |
| --- | --- | --- |

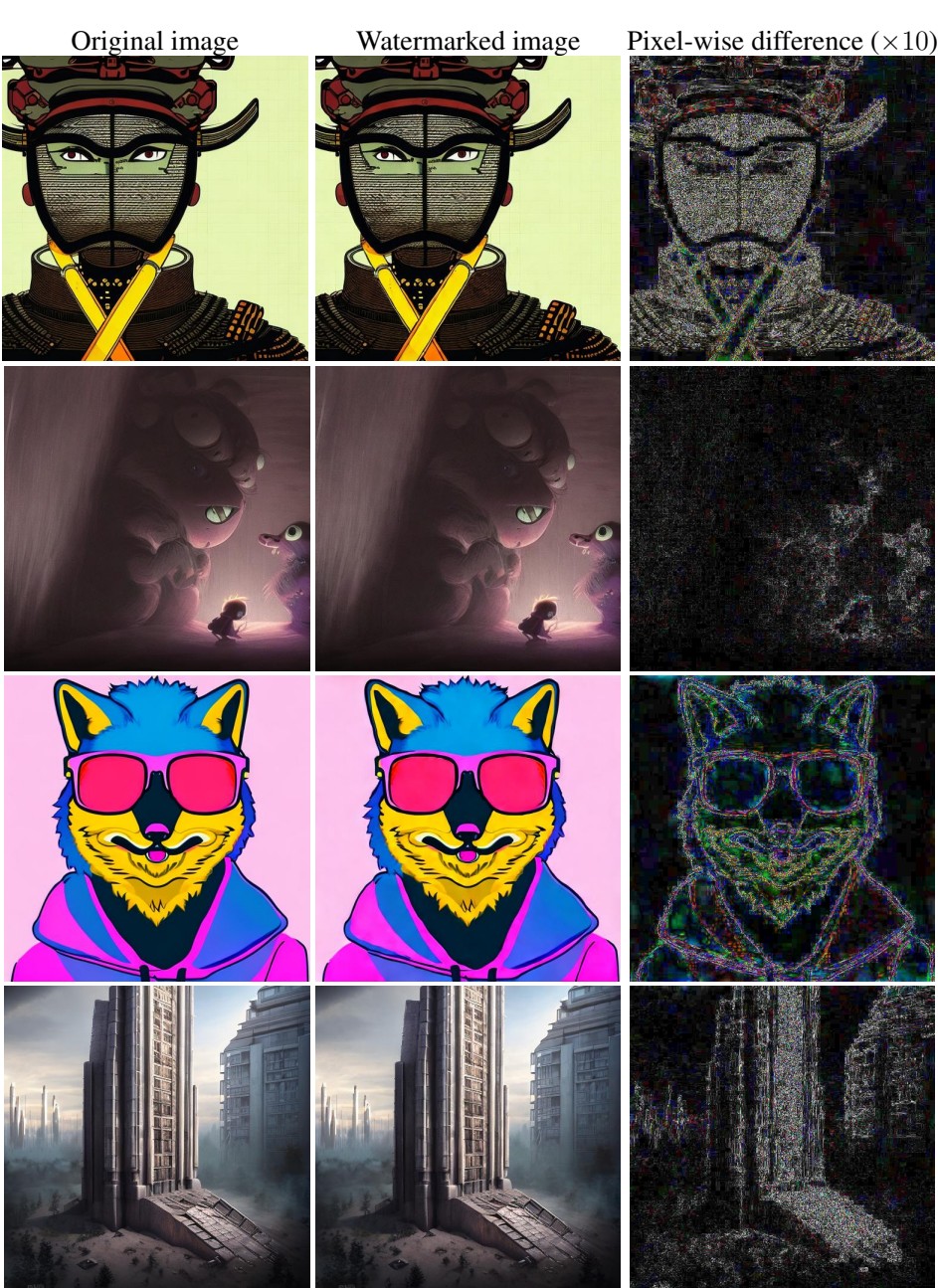

Figure 7: Examples of images generated via inserting the watermark in a Fourier invariant.

