# OpenReview forum: "Spread them Apart: Towards Robust Watermarking of Generated Content"
_ICLR.cc/2025/Conference — ICLR 2025 Conference Withdrawn Submission_

### Official Review · Reviewer_C9YY · 2024-10-29

**Soundness:** 2
**Presentation:** 2
**Contribution:** 2
**Rating:** 5
**Confidence:** 5

**Summary:**

A framework called “Spread them Apart” for watermarking generated content is presented, specifically targeting images produced by diffusion models.

**Strengths:**

+ A method of embedding watermarks during the model inference phase is proposed to simultaneously detect and attribute generated images.
+ Proven robustness against bounded additive perturbations.

**Weaknesses:**

- The expression and logic of the paper can be further improved to enhance readability. For instance, a clear explanation of "$\mathcal{L_{qual}}$​" is lacking. Additionally, there are a few typos, such as the use of "the" in line 115.

- To enhance the understanding of the paper's innovation, the authors should compare their approach with using pixel differencing in digital watermarking [1]. By highlighting the differences between the two methods, the paper can further illustrate the unique contribution and novelty of the proposed approach.

- While resilience against certain "watermark removal" attacks is demonstrated, this paper lacks consideration of the effectiveness of the proposed watermarking method against ambiguity attacks.

- The evaluation is not very convincing. It is best to supplement some experiments that include the latest methods, such as WOUAF [2].

  [1] A Robust and Computationally Efficient Digital Watermarking Technique Using Inter Block Pixel Differencing.

  [2] WOUAF: Weight Modulation for User Attribution and Fingerprinting in Text-to-Image Diffusion Models.

**Questions:**

- The method presented in this paper cannot provide robustness against transformation attacks such as cropping and rotation. The authors claim that embedding watermarks in a local area can overcome this problem. How is this achieved?
- Can the proposed method resist watermark overwriting attacks?

---

> ### Author Response · Authors · 2024-11-24
> **Answer to Reviewer C9YY**
>
> ### Thank you for your valuable feedback. In our response below, we address the weaknesses you indicate and answer your questions.
>
>
> ## W1.
>
> Thank you for the careful reading. We have fixed several typos in the updated manuscript and clarified our notations, where needed.
>
> ## W2.
> The paper you mention is very relevant to our work. However, the method described is more about computing the fingerprint of the particular image (namely, when the image is ready, one can compute the fingerprint, making it infeasible, for example, for the attribution problem); in contrast, in our method, an optimization is performed to obtain an image with the predefined watermark.
>
> ## W3 and Q2.
>
> In the manuscript, we consider two types of ambiguity attacks: (i) unintentional ones, occurring because of the large number of users in the database, and (ii) adversarial ones, where an attacker aims to erase an embedded watermark.
>
> We report the robustness to attack of the first type by depicting TPR in the attribution problem (please see Table below).
>
> Table: TPR given different values of m of number of users in the database.
> | m | TPR@FPR=10^-6 |
> | :---------------- | :------:
> | 10    | 1.00 |
> | 1000    | 1.00 |
> | 10000    | 1.00 |
> | 1000000    | 1.00 |
> | 10000000    | 1.00 |
> | 100000000    | 1.00 |
>
> The robustness to the second type of attacks is demonstrated by the evaluation of TPR against PGD attack, aimed at erasing the watermark. Please see one in the Tables (3-4) in the manuscript.
>
> Please note that to perform a watermark overwriting attack, an attacker has to have a white-box  access to the generation and watermarking pipeline; in our work, we do not consider such access.
>
> ## W4.
>
> Following your suggestion, we have added more relevant work for the comparison (please see tables 2-3-4 in the main text and table 9 in the appendix).
>
> ## Q1.
>
> It is noteworthy that slight modifications to our approach can be made to provide robustness to rotations and translations. Namely, one can embed watermarks in the invariants in the Fourier space (see Theorem 6 for rotations and Theorem 3 for translations from [1]).
>
> When the watermarks are embedded, due to invariance from above, they are guaranteed to retain when the corresponding transformation is applied. We present the results in the updated version of the appendix. And report them here, in Table 1. Note that the invariance to rotations in general can not be guaranteed in practice due to interpolation errors.
>
>
>
> Table 1. TPRs under geometric transformations, JPEG, cropping and erasing,  detection problem. We fix FPR = 10^-6
> | Method              | Rot(10)  | Translation (30 X 30) | JPEG(50) | Crop(400 X 400) | Erase(50 X 50) |
> | :---------------- | :------: | ----: |:--------------------: |  :-------:   | :------: |
> | Ours (Fourier)      |  0.85 | 1.00 | 0.70 |  0.80   | 0.90 |
> | Stable sign.         |  0.97   | - | 0.88 |  0.98   | - |
> | SSL   |   -   | - | 0.97 |  1.00   | - |
> | AquaLora |  1.00   | - | 0.99 |  0.91   | - |
> | WOUAF |  0.99   | - |0.97 |  0.98   | 0.99 |
>
>
>
> We evaluate our method against cropping and image erasing when the watermark is embedded in the invariant to translations (Theorem 3 from [1]).
>
>
>
> ## References:
> [1] Lin, Feng, and Robert D. Brandt. "Towards absolute invariants of images under translation, rotation, and dilation." Pattern Recognition Letters 14.5 (1993): 369-379.

---

> > ### Comment · Reviewer_C9YY · 2024-11-26
> > **Official Comment by Reviewer  C9YY**
> >
> > Thanks for your response. It seems that the robustness of the proposed method is not signifantly better than that of the  current method. Therefore, I remain my score.

---

### Official Review · Reviewer_yiE4 · 2024-10-31

**Soundness:** 2
**Presentation:** 1
**Contribution:** 2
**Rating:** 3
**Confidence:** 3

**Summary:**

The paper introduces an image watermark method named spread them apart. When generating images from latent $z$, the model measures the generated image to see if it satisfies the constraint $L_{wm} < \epsilon$. If the watermark doesn’t successfully embed into the image, the latent $z$ is optimized, otherwise, the image is given to the user. To decide on the threshold, the method uses double-tail detector as described [1]. To embed the watermark, the method generate a unique secret $s(u_i)$ and a watermark $w(u_i)$, then embed it on the pixel level. If the embedding is not successful, the method further fine-tunes the latent vector $z$, with a two-component loss. The paper also provides a proof of robustness against additive attacks. The experiment shows that the method demonstrates strong robustness against various watermarking removal attacks, while still maintaining high image quality.


[1] Zhengyuan Jiang, Jinghuai Zhang, and Neil Zhenqiang Gong. 2023. Evading Watermark based Detection of AI-Generated Content. In Proceedings of the 2023 ACM SIGSAC Conference on Computer and Communications Security (CCS '23). Association for Computing Machinery, New York, NY, USA, 1168–1181. https://doi.org/10.1145/3576915.3623189

**Strengths:**

1. Compared with stable signature [1] and SSL [2], the method demonstrates strong robustness against attacks like brightness, contrast shift, gamma correction, sharpening, hue, satuation adjustment, noise attack, JPEG and PGD [3].
2. The paper is well-written, and has a clear structure.

[1] Fernandez, Pierre, et al. "The stable signature: Rooting watermarks in latent diffusion models." Proceedings of the IEEE/CVF International Conference on Computer Vision. 2023.

[2] Fernandez, Pierre, et al. "Watermarking images in self-supervised latent spaces." ICASSP 2022-2022 IEEE International Conference on Acoustics, Speech and Signal Processing (ICASSP). IEEE, 2022.

[3] Mądry, Aleksander, et al. "Towards deep learning models resistant to adversarial attacks." stat 1050.9 (2017).

**Weaknesses:**

1. The paper does not include ablation study, which is a large missing. For example, you could present the impact of different watermark length, the effect of epsilon parameter to the robustness nad visibility, the loss weight ($\lambda_{wm}$ and $\lambda_{qual}$) to the robustness and visibility.
2. The method embeds watermark in the pixel space. The latent vector $z$ is optimized during inference to constraint the loss within the acceptable region . It seems all models that have a decoder structure can perform this method, which is pretty general, like VAE or GAN. Could the authors explain the reason why only include experiments on diffusion model? Or could the authors suggest the possible disadvantages of other types of models? Or the authors can pointing out the property that the diffusion model adapted well on their proposed method.
3. The experiment only includes stable signature and SSL as baselines, which is not comprehensively compared. Can the authors include more baselines like WOUAF [1]?
4. The optimization process introduces more computation during inference. The authors are encouraged to report the extra time used in the optimization. For example, the authors can present hte average inference time with and without watermarking, or the optimization time scales with watermark length. You could also present your computation overhead compared to other watermarking methods (baselines).

[1] Kim, C., Min, K., Patel, M., Cheng, S., & Yang, Y. (2024). Wouaf: Weight modulation for user attribution and fingerprinting in text-to-image diffusion models. In Proceedings of the IEEE/CVF Conference on Computer Vision and Pattern Recognition (pp. 8974-8983).

**Questions:**

1. The method embeds and extracts the watermark in a pixel-wise manner. A single pixel can be easily altered by attackers. Could you consider applying your method to block-wise areas, similar to the approach in [1]?

[1] Yang, Z., Zeng, K., Chen, K., Fang, H., Zhang, W., & Yu, N. (2024). Gaussian Shading: Provable Performance-Lossless Image Watermarking for Diffusion Models. In Proceedings of the IEEE/CVF Conference on Computer Vision and Pattern Recognition (pp. 12162-12171).

---

> ### Author Response · Authors · 2024-11-24
> **Answer to Reviewer yiE4**
>
> ### Thank you for your valuable feedback. In our response below, we address the weaknesses you indicate and answer your questions.
>
>
> ## W1.
>
> We have added the ablation study on the hyperparameters you proposed, namely, watermark length, epsilon, and loss weights. In terms of the trade-off between attack resilience and perceptual quality of the generated images (Table 1-2 here), we see that the default parameters are close to the optimal ones. We report the results of the ablation study in the appendix, Section A.1.3.
>
> Table 1. Ablation study: the effect of the parameter values on the robustness of the watermark. We report an average bit-wise error and study the robustness to JPEG, Hue, Saturation, Sharpness and Gaussian noise, since our approach provide robustness to brightness, contrast and gamma shifts by design. In the "Parameter" column, we report the varying parameter; the other parameters are set to default values (n=50, eps=0.2, `\lambda_wm`=0.9, `\lambda_qual`=150)
>
> | Parameter | Value | JPEG | Hue | Saturation | Sharpness | Noise |
> | :---------------- | :------: | :----: |:--------------: |  :-------:   | :------: | :------:|
> |n|50|0.123|0.013| 0.095|0.002|0.049|
> |n|100|0.143|0.011|0.104|0.001|0.056|
> |n|150| 0.157|0.013|0.112|0.001|0.063|
> |n|250|0.159|0.015|0.120|0.001|0.069|
> --
> |eps| 0.0|0.313|0.109|0.206|0.016|0.202|
> |eps|0.05|0.261|0.055|0.169|0.005|0.159|
> |eps|0.2|0.143|0.011|0.104|0.001|0.056|
> |eps|0.5|0.054|0.001|0.041|0.000|0.003|
> --
> |`lambda_wm`|0.5|0.150|0.015|0.108|0.002|0.060|
> |`lambda_wm`|0.9|0.143|0.011|0.104|0.001|0.056|
> |`lambda_wm`|2.0|0.136|0.012|0.103|0.001|0.056|
> --
> |`lambda_qual`|10.0|0.059|0.014|0.071|0.004|0.035|
> |`lambda_qual`| 50.0|0.088|0.008|0.082|0.001|0.040|
> |`lambda_qual`| 150.0|0.143|0.011|0.104|0.001|0.056|
> |`lambda_qual`|200.0|0.160|0.013|0.109|0.001|0.060|
> -----
>
> Table 2. Ablation study: the effect of the parameter values on the image quality. We report the values of SSIM, PSNR, LPIPS  image quality metrics.
>
> | Parameter  | Value | SSIM | PSNR | LPIPS|
> | ------------- |:-------------:|:-------------:|:-------------:|:-------------:|
> | n      |50|0.897|31.104|0.006|
> | n      |100|0.856|29.381|0.007|
> | n      |150|0.827|28.309|0.009|
> | n      |250|0.777|26.726|0.013|
> |eps|0.0|0.878|30.142|0.006|
> |eps|0.05|0.873|29.937|0.007|
> |eps|0.2|0.856|29.381|0.007|
> |eps|0.5|0.820|28.378|0.010|
> | `lambda_wm`      |0.5|0.869|29.830|0.006|
> | `lambda_wm`     |0.9|0.856|29.381|0.007|
> | `lambda_wm`     |2.0|0.842|28.912|0.008|
> | `lambda_qual`     |10.0|0.752|26.200|0.057|
> | `lambda_qual`      |50.0|0.806|27.601|0.019|
> | `lambda_qual`      |150.0|0.856|29.381|0.007|
> | `lambda_qual`      |200.0|0.869|29.918|0.005|
>
>
>
> ## W2.
>
> Indeed, our approach is not limited to the diffusion models and, in principle, can be applied to any decoder-based model. We have chosen to stick to DM, since the quality of images generated by the diffusion models significantly surpasses that of VAE and GANs. Therefore, there is a much greater interest in their protection and safe deployment.
>
> ## W3.
> Following your suggestion, we have included more baselines to compare our approach against (Please see Tables 2-3-4 in the main text and Table 9 in the Appendix).
>
> ## W4.
> Please find the comparison with the other approaches in terms of time required to embed a watermark in the Table 3 below.
>
> Table 3: Average time in seconds required to embed a watermark.
>
> |Method | Time, seconds|
> |:-----:|:-----:|
> |Ours| 36.7 |
> |Stable sign | ~ 60.0|
> |SSL | - |
> |AquLora | ~ 0.0|
> |WOUAF | 1.1 |
>
>
> ## Q1.
> Thank you for the idea on the possible enhancement of our method. The paper that you have shared with us inserts a watermark during the sampling process, therefore we cannot benefit from this idea straightforwardly.
> Regarding the block-wise watermark insertion in our setting per se, likely the perceptual image quality will degrade, because more pixels will be involved in the watermark insertion, leaving artifacts in an image. In such a case, more restrictions on pixel values should be added, and it needs further research and pondering.
>
>
>
>
> ## References:
> [1] Lin, Feng, and Robert D. Brandt. "Towards absolute invariants of images under translation, rotation, and dilation." Pattern Recognition Letters 14.5 (1993): 369-379.

---

> ### Comment · Reviewer_yiE4 · 2024-11-26
>
> I appreciate authors for their efforts in this work. After reconsideration, I can’t raise my score. Table 1 in the paper indicates that the image consistency of the proposed method is worse than others in most metrics. Table 5 indicates that the time required in the watermark embedding is high. Also, the authors’ reply to W2 indicates that their understanding of diffusion model is limited. Owing to these factors, I will keep my score.
>
> Here are some suggestions for authors. Your method can embed high-capacity watermarks. It would be better to demonstrate them in the experiment. Also, your method is training-free, which should also be emphasized. I guess there is a trade-off between the image quality and the watermark capacity, so make sure you achieve the balance. And finally please use different color texts for the revision, which makes it more clear for reviewers to know the difference you made with respect to the previous version.
>
> Also, I can’t understand why in many tables you use a horizontal line where there’s supposed to be a number, for example the SSIM and PSNR for WOUAF, and the Hue, Saturation for AquaLora. Can the authors explain?

---

> > ### Author Response · Authors · 2024-11-27
> > **official comment from the authors**
> >
> > We thank you for the suggestions also provided by reviewer MwGz. Regarding the answer to W2 (which only motivates to stick to diffusion models in the assessment of the proposed approach): it is pretty confusing how any conclusion about the knowledge of DM can be made from it.

---

### Official Review · Reviewer_uwy1 · 2024-11-02

**Soundness:** 2
**Presentation:** 3
**Contribution:** 2
**Rating:** 3
**Confidence:** 5

**Summary:**

The paper "Spread them Apart" introduces a robust watermarking technique for images generated by diffusion models, embedding watermarks directly during the generation process to prevent unauthorized removal.

**Strengths:**

1. The watermarking method enables accurate identification of the user who sent the query, ensuring traceability.
2. The watermark minimally alters the image, maintaining high visual quality while embedding robust identifiers.

**Weaknesses:**

1. The robustness evaluation lacks common post-generation attacks like rotation, resizing, grayscale conversion, and cropping, and examples of corrupted images are not provided.
2. The watermark embedding process increases the image generation time. The paper should provide the experiments of generation time cost.
3. Experiments assessing the impact of watermarking on image quality are limited. More quantitive metrics like PSNR should be used

**Questions:**

See weakness

---

> ### Author Response · Authors · 2024-11-24
> **answer to Reviewer uwy1**
>
> ## Thank you for your valuable feedback. In our response below, we address the weaknesses you indicate, answer your questions and provide results of additional experiments, where requested.
>
>
> ### W1.
> We agree that our method does not provide robustness to post-generation attacks as is. However, slight modifications to our approach can be made to provide robustness to rotations and translations. Namely, one can embed watermarks in the invariants in the Fourier space (see Theorem 6 for rotations and Theorem 3 for translations from [1]).
>
> When the watermarks are embedded, due to invariance from above, they are guaranteed to retain when the corresponding transformation is applied. We present the results in the updated version of the appendix. And report them here, in Table 1. Note that the invariance to rotations in general can not be guaranteed in practice due to interpolation errors.
>
>
>
> Table 1. TPRs under geometric transformations, JPEG, cropping and erasing,  detection problem. We fix FPR = 10<sup>-6
> | Method              | Rot(10)  | Translation (30 X 30) | JPEG(50) | Crop(400 X 400) | Erase(50 X 50) |
> | :---------------- | :------: | ----: |:--------------------: |  :-------:   | :------: |
> | Ours (Fourier)      |  0.85 | 1.00 | 0.70 |  0.80   | 0.90 |
> | Stable sign.         |  0.97   | - | 0.88 |  0.98   | - |
> | SSL   |   -   | - | 0.97 |  1.00   | - |
> | AquaLora |  1.00   | - | 0.99 |  0.91   | - |
> | WOUAF |  0.99   | - |0.97 |  0.98   | 0.99 |
>
>
>
> We evaluate our method against cropping and image erasing when the watermark is embedded in the invariant to translations (Theorem 3 from [1]).
>
>
> ### W2.
> Please see the results in Table 2 below.
>
>
> Table 2: Average time in seconds required to embed a watermark.
>
> |Method | Time, seconds|
> |:-----:|:-----:|
> |Ours| 36.7 |
> |Stable sign | ~ 60.0|
> |SSL | - |
> |AquLora | ~ 0.0|
> |WOUAF | 1.1 |
>
> It is noteworthy that our method allows to extract the watermark in  less than 1 second when the number of users is 100000.
>
>
> ## W3.
> Please note that the quality metrics like PSNR were reported in the original manuscript. (Table 1). In the updated version, we add the comparison with other baselines in terms of the computation cost.
>
>
> ## References:
> [1] Lin, Feng, and Robert D. Brandt. "Towards absolute invariants of images under translation, rotation, and dilation." Pattern Recognition Letters 14.5 (1993): 369-379.

---

> > ### Comment · Reviewer_uwy1 · 2024-11-24
> >
> > Thanks for your reply!
> >
> > I will keep my score because the results in Table 1 and Table 2 indicate that the robustness is worse than others in most of image corruption and the time cost is also high than other methods.

---

### Official Review · Reviewer_uXyP · 2024-11-03

**Soundness:** 3
**Presentation:** 3
**Contribution:** 3
**Rating:** 5
**Confidence:** 4

**Summary:**

This paper introduces "Spread Them Apart", an in-processing watermarking algorithm that aims to achieve robust image attribution. The idea is to randomly generate n index pairs (A_i, B_i). By comparing the pixel intensities at these indexed locations, a tuple of n binary keys is established, based on the relative magnitudes of the pixel values in each pair. This tuple can then serve as a unique, verifiable identifier for the user. The proposed method demonstrates robustness against additive noise attacks within a bounded magnitude, ensuring the integrity of the watermark under various perturbations.

**Strengths:**

This paper is well-written and easy to follow. Although the methodology itself is straightforward, it is proven to be effective and robust against simple image manipulation.

**Weaknesses:**

The approach presented in this paper is relatively straightforward, demonstrating theoretical robustness against perturbations such as brightness adjustment, contrast shifts, and additive noise. However, its limitations are also apparent. As the authors acknowledge, the method lacks resilience against geometric distortions, which alter the image's size and indexing—common forms of attacks that the paper does not address in detail.

While the authors suggest that "this limitation can be addressed by embedding watermarks in localized areas or the frequency domain of the image," this raises the question: given these potential solutions, why these methods are not implemented in the current study? It would be insightful to understand the trade-offs here.

A further concern is the scalability of the proposed approach. The watermark retrieval process requires comparing each image against every user's secret key before generating a binary string, which appears impractical as the user base grows. This limitation suggests that, as the number of users increases, the approach may face significant challenges in maintaining efficiency and scalability.

**Questions:**

1. In this paper, there are some constraints set in place on the attacks (i.e. L infinite norm for additive noise, etc.) To evaluate the effectiveness of these constraints, it would be helpful to include visualizations of the attacked images in the appendix. Such visual aids could convincingly illustrate the impact of these constrained attacks on image quality and watermark robustness.

2. For the "contrast -" result in Table 2, it is 0.998. This value is influenced by a negative sign introduced during processing. Given that the method employs a two-tailed detection approach, this result is indeed the optimal outcome in this context. However, a brief explanatory note could help clarify this for readers, potentially reducing confusion about the significance of the negative sign and its impact on the results.

---

> ### Author Response · Authors · 2024-11-24
> **Answer to Reviewer uXyP**
>
> ## Thank you for your valuable feedback. In our response below, we address the weaknesses you indicate, answer your questions and provide results of additional experiments, where requested.
>
>
> ### W1 and W2.
> We agree that our method does not provide robustness to rotations, translations, and cropping as is. However, slight modifications to our approach can be made to provide robustness to rotations and translations. Namely, one can embed watermarks in the invariants in the Fourier space (see Theorem 6 for rotations and Theorem 3 for translations from [1]).
>
> When the watermarks are embedded, due to invariance from above, they are guaranteed to retain when the corresponding transformation is applied. We present the results in the updated version of the appendix. And report them here, in Table 1. Note that the invariance to rotations in general can not be guaranteed in practice due to interpolation errors.
>
>
>
> Table 1. TPRs under geometric transformations, JPEG, cropping and erasing,  detection problem. We fix FPR = 10^-6
> | Method              | Rot(10)  | Translation (30 X 30) | JPEG(50) | Crop(400 X 400) | Erase(50 X 50) |
> | :---------------- | :------: | ----: |:--------------------: |  :-------:   | :------: |
> | Ours (Fourier)      |  0.85 | 1.00 | 0.70 |  0.80   | 0.90 |
> | Stable sign.         |  0.97   | - | 0.88 |  0.98   | - |
> | SSL   |   -   | - | 0.97 |  1.00   | - |
> | AquaLora |  1.00   | - | 0.99 |  0.91   | - |
> | WOUAF |  0.99   | - |0.97 |  0.98   | 0.99 |
>
>
>
> We evaluate our method against cropping and image erasing when the watermark is embedded in the invariant to translations (Theorem 3 from [1]).
>
>
> ### W3.
> Regarding the scalability of the approach: we have evaluated the time cost required to extract the watermark and compare it to the public keys of the users in the database. Please see the results in Table 2 below.
>
>
>
> Table 2: Average time in seconds required to extract a watermark, depending on the number m of users.
>
> |m | Time, seconds|
> |:-----:|:-----:|
> |1| 7.5x10^-5 |
> |10 | 7.4x10^-4|
> |1000| 7.2x10^-2 |
> |10000 | 6.9x10^-1|
> |1000000 | 71.2 |
>
> It is noteworthy that our method allows to extract the watermark in  less than 1 second when the number of users is 100000.
>
>
>
> ### Q1.
> Following your suggestion, we have added examples of corrupted images in the manuscript (please see the appendix, Fig 6)
>
> ### Q2.
>
> Following you suggestion, we have added explanation for the transform "Contrast -" in the manuscript (specifically, we indicated how it  differs from "Contrast +").
>
>
>
> ## References:
> [1] Lin, Feng, and Robert D. Brandt. "Towards absolute invariants of images under translation, rotation, and dilation." Pattern Recognition Letters 14.5 (1993): 369-379.

---

> > ### Comment · Reviewer_uXyP · 2024-11-25
> >
> > I appreciate the author for their effort on additional experiment results. However, I do have some additional concerns. I understand that embedding in the Fourier space can indeed provide some robustness against geometric distortion, however, in my experience, there may come a price on image quality. Please correct me if I'm wrong but I do not see any specific mechanism in the proposed method that can neglect the image quality difference between embedding in pixel domain or frequency domain. So there exists a potential trade-off between robustness in geometric distortion and image quality when frequency domain embedding is implemented. Since this trade-off is still unclear to me, the robustness results in geometric distortion are not very informative at the paper's current stage. So I will keep my current score unless further clarification can be made.

---

> > > ### Author Response · Authors · 2024-11-26
> > > **Official comment by the Authors**
> > >
> > > Indeed, embedding the watermark in the Fourier space can lead to the degradation of the image quality. However, in our approach, embedding is done by minimizing the loss function that controls both the watermarking process and the deviation in quality from an unwatermarked image; hence, we can maintain the image quality. Please see Fig. 7 in the updated appendix for the examples of images watermarked in the Fourier space (visually, they can be even better than those watermarked in the pixel space).

---

### Official Review · Reviewer_MwGz · 2024-11-04

**Soundness:** 3
**Presentation:** 3
**Contribution:** 3
**Rating:** 6
**Confidence:** 4

**Summary:**

This paper proposed to watermark the images generated by latent diffusion models, for detection and attribution, where the watermark is embedded by optimizing the denoised latent representation passed to the VAE decoder of the latent diffusion model. The evaluation of the proposed watermarking method focuses on the robustness against different watermark removal attacks.

**Strengths:**

1. The proposed watermarking method is shown to be robust against various types of removal attacks, by conducting comprehensive experiments.
2. The bound of the robustness against additive watermark removal attacks is theoretically analyzed.
3. The paper is well-written overall.

**Weaknesses:**

1. The paper does not provide the evaluation of robustness against cropping, rotation, and translation attacks.
2. The proposed method needs 700 iterations to optimize the latent representation before generating an image, slowing down the generation speed.
3. More relevant watermarking methods should be included for comparison, for example, AquaLoRA [1].

[1] Feng, Weitao, et al. "AquaLoRA: Toward White-box Protection for Customized Stable Diffusion Models via Watermark LoRA." arXiv preprint arXiv:2405.11135 (2024).

**Questions:**

1. Compared with the original inference process, what is the additional time cost introduced by the extra optimization process of the denoised latent vector for watermarking?
2. The paper claims that “The watermarking method does not provide robustness against cropping, rotation, and translation attacks. However, this limitation can be overcome by inserting watermarks in the localized areas or the frequency domain of the image”. Why inserting watermarks in the localized areas or the frequency domain of the image can improve the robustness against cropping, rotation, and translation attacks?
3. Is the proposed method robust to image editing attacks?

---

> ### Author Response · Authors · 2024-11-24
> **Author response to Reviewer MwGz**
>
> ### Thank you for your valuable feedback. In our response below, we address the weaknesses you indicate and answer your questions.
>
>
> ## W1 and Q2.
>
> Indeed, our method does not provide robustness to rotations, translations, and cropping. However, slight modifications to our approach can be made to provide robustness to rotations and translations. Namely, one can embed watermarks in the invariants in the Fourier space (see Theorem 6 for rotations and Theorem 3 for translations from [1]).
>
> When the watermarks are embedded, due to invariance from above, they are guaranteed to retain when the corresponding transformation is applied. We present the results in the updated version of the appendix. Note that the invariance to rotations in general can not be guaranteed in practice due to interpolation errors.
>
>
>
> Table 1. TPRs under geometric transformations, JPEG, cropping and erasing,  detection problem. We fix FPR = 10^-6
> | Method              | Rot(10)  | Translation (30 X 30) | JPEG(50) | Crop(400 X 400) | Erase(50 X 50) |
> | :---------------- | :------: | ----: |:--------------------: |  :-------:   | :------: |
> | Ours (Fourier)      |  0.85 | 1.00 | 0.70 |  0.80   | 0.90 |
> | Stable sign.         |  0.97   | - | 0.88 |  0.98   | - |
> | SSL   |   -   | - | 0.97 |  1.00   | - |
> | AquaLora |  1.00   | - | 0.99 |  0.91   | - |
> | WOUAF |  0.99   | - |0.97 |  0.98   | 0.99 |
>
>
>
> We evaluate our method against cropping and image erasing when the watermark is embedded in the invariant to translations (Theorem 3 from [1]).
>
>
> ## W2  and Q1.
> Please find the comparison with the other approaches in terms of time required to embed a watermark in the Table 2 below.
>
>
> Table 2: Average time in seconds required to embed a watermark.
>
> |Method | Time, seconds|
> |:-----:|:-----:|
> |Ours| 36.7 |
> |Stable sign | ~ 60.0|
> |SSL | - |
> |AquLora | ~ 0.0|
> |WOUAF | 1.1 |
>
>
> ## Q3.
> Among image editing attacks, we consider image erasing (please see the results in Table 1 here and in Table 9 in the appendix).
>
>
>
>
> ## References:
> [1] Lin, Feng, and Robert D. Brandt. "Towards absolute invariants of images under translation, rotation, and dilation." Pattern Recognition Letters 14.5 (1993): 369-379.

---

> ### Comment · Reviewer_MwGz · 2024-11-26
>
> Thanks for addressing my concerns and I appreciate your effort in conducting additional experiments. I want to keep my rating. Here are two more suggestions: 1) You have mentioned that the proposed method embeds 100 bits long watermarks, while the competitors (Stable Sign., AquaLoRA, WOUAF) embed much shorter (at most 48 bits) watermarks in your experiments. I think this information about the embedding capacity (which is an advantage of your method) should also appear in the table. 2) Your method does not need additional training of generative model, while the competitors (Stable Sign., AquaLoRA, WOUAF) do, which I think is another important piece of information that should appear in your table.

---

> > ### Author Response · Authors · 2024-11-27
> > **Official comment from authors**
> >
> > We thank you for the important suggestions, we will include them in the updated version of the manuscript.

---

### Note · Authors · 2024-11-27

I have read and agree with the venue's withdrawal policy on behalf of myself and my co-authors.